# Treatment of Parkinson’s Disease with Cognitive Impairment: Current Approaches and Future Directions

**DOI:** 10.3390/bs11040054

**Published:** 2021-04-17

**Authors:** Chichun Sun, Melissa J. Armstrong

**Affiliations:** Department of Neurology, University of Florida, Gainesville, FL 32611, USA; chichun.sun@neurology.ufl.edu

**Keywords:** Parkinson’s disease, mild cognitive impairment, dementia, treatment

## Abstract

Cognitive impairment risk in Parkinson’s disease increases with disease progression and poses a significant burden to the patients, their families and society. There are no disease-modifying therapies or preventative measures for Parkinson’s disease mild cognitive impairment (PD-MCI), or Parkinson’s disease dementia (PDD). This article reviews current and previously investigated treatments and those under investigation, including pharmacologic, non-pharmacologic and surgical procedures. There are currently no effective pharmacologic or non-pharmacologic treatments for PD-MCI. The only recommended treatment for PDD currently is rivastigmine, a cholinesterase inhibitor. Donepezil and galantamine—other cholinesterase inhibitors—are possibly useful. Memantine, a N-methyl-D-aspartate (NMDA) receptor antagonist, is considered investigational in PDD. Drug repurposing (atomoxetine, levodopa, insulin, atomoxetine for PD-MCI; ambroxol and ceftriaxone for PDD) and novel medications (SYN120, GRF6021, NYX-458 for PD-MCI; ANAVEX2-73, LY3154207, ENT-01, DAAOI-P for PDD) currently have insufficient evidence. There is growing research supporting exercise in the treatment of PD-MCI, but most non-pharmacological approaches have insufficient evidence for use in PD-MCI (cognitive rehabilitation, deep brain stimulation, transcranial direct current stimulation, transcranial ultrasound, vestibular nerve stimulation) and PDD (cognitive intervention, deep brain stimulation, transcranial alternating current stimulation, transcranial ultrasound, temporal blood brain barrier disruption). Research is needed for both disease-modifying and symptomatic treatments in PD cognitive impairment.

## 1. Introduction

Parkinson’s disease (PD) was historically classified as a movement disorder, but cognitive impairment is prevalent, especially later in the disease process. PD-related cognitive impairment is divided into two categories depending on severity and whether the level of cognitive impairment interferes with activities of daily living (ADLs). In PD mild cognitive impairment (PD-MCI), there is no impairment in ADLs. In PD dementia (PDD), ADLs are affected by cognition. The goal of this narrative review is to describe the therapeutic approaches studied for both PD-MCI and PDD, highlight the approaches recommended by the Movement Disorders Society, and provide a brief overview of therapies actively under investigation for treating cognitive symptoms in PD. To provide context for current and experimental approaches, we first briefly review the epidemiology and diagnosis of PD-MCI and PDD and the proposed mechanisms for cognitive impairment in PD.

### 1.1. Epidemiology and Clinical Significance

PD-MCI is present in 25–30% of persons with PD without dementia [1] and it increases the relative risk of dementia compared to those with PD without MCI (relative risk of 39.2 at 3 years) [2]. At 5 years, the risk of dementia is 6.5 times higher in individuals with PD-MCI, independent of sex, age, years of education and motor function [3]. Dementia is present in 24–31% of persons with PD [2,4]. The risk of dementia in persons with PD is 4 to 6 times higher than healthy individuals matched for age, sex, and education [5]. At least 75% of persons with PD who survive more than 10 years develop dementia [5]. PD-MCI and PDD are associated with decreased quality of life for persons with PD and caregivers and also increased nursing home admissions [6].

### 1.2. Diagnosis

Diagnoses of PD-MCI and PDD rely on the history provided by the patients and their caregivers, as well as cognitive testing. In contrast to Alzheimer’s disease (AD) dementia, the primary cognitive domains affected in PD are the executive, attentional and visuospatial domains [1]. The 2012 PD-MCI diagnostic criteria consist of level I (abbreviated) and level II (comprehensive) assessments [7]. For a diagnosis of PD-MCI, individuals must have a diagnosis of idiopathic PD, a gradual decline in cognitive abilities (noted by the patient, informant, or clinician), cognitive deficits upon testing, and preserved functional independence (with the exception of subtle difficulties with complex tasks). Level I assessment requires impairment on a scale of global cognition (e.g., the Montreal Cognitive Assessment (MoCA)) or at least two neuropsychological tests. Level II assessment requires full neuropsychological testing that includes all five cognitive domains, with impairment on two tests within one cognitive domain or on one test in two cognitive domains. The 2007 Movement Disorders Society PDD diagnostic criteria require a diagnosis of PD, the insidious onset and slow progression of a dementia syndrome (with impairment in more than one cognitive domain), decline from a premorbid level, and impaired ADLs beyond those attributed to motor and autonomic symptoms [8].

### 1.3. Risk Factors

The most consistently reported risk factors for PDD include more severe parkinsonism, increased age, and MCI [5], as well as other factors, including hallucinations, speech impairment, low level of education, depression and male sex [9]. Cognitive domains associated with higher risk of conversion from PD-MCI to PDD include poor baseline episodic memory, visuospatial function, semantic verbal fluency and cognitive flexibility [3].

### 1.4. Pathophysiology

The underlying mechanism of cognitive impairment in PD is not well understood and likely includes multiple factors. Proposed contributors include abnormal protein deposition, loss of dopaminergic neurons, neurotransmitter deficits, synaptic dysfunction, genetics [1], fatty acid oxidation [10], inflammation and oxidative stress [11,12,13], exosomal dysfunction [14], the gut microbiome and gut–brain axis involving the autonomic and enteric nervous systems [13], and prion-like aggregation of alpha-synuclein [13]. Many of these processes are also involved in other neurodegenerative diseases. The Braak hypothesis posits that aberrant alpha-synuclein starts in the gut and ascends via the vagus nerve to the brain, and that intraneuronal deposits made of α-synuclein fibrils (Lewy bodies) deposit in a caudal–cranial fashion, starting with the dorsal nucleus of the vagus and olfactory bulbs to the substantia nigra, then limbic, paralimbic and associative cortices [15]. The later stages are thought to cause cognitive dysfunctions in PD. Multiple abnormal proteins have been found in the postmortem brains of individuals with PD, including α-synuclein, tau and amyloid [11]. Individuals with PDD can have AD pathology, characterized by low CSF levels of amyloid β-42 (Aβ-42) and the cortical deposition of Aβ-42 in neuroimaging [11]. Individuals with dual PD and AD neuropathologies have more rapid disease progression (mean survival of 4.5 years, versus 10 years in those without AD neuropathology) [16]. This overlapping pathology presents as a diagnostic challenge in evaluating individuals with cognitive complaints. With regard to the gut microbiome, a significant alteration in gut bacterial species was found in individuals with PD-MCI as compared to PD with normal cognition and healthy controls [17]. Certain species correlated with worsening cognition.

Neurotransmitters implicated in cognitive changes and psychosis in PD include dopamine, serotonin, norepinephrine, acetylcholine, and N-methyl-D-aspartate (NMDA), prompting exploration of all of these targets for therapeutic drug development (Table 1). Additionally, several potential genetic risk factors have been identified for cognitive impairment in PD, including GBA (glucosylceremidase), MAPT (microtubule-associated protein tau), APOE (apolipoprotein E), LRRK2 (leucine-rich repeat serine/threonine-protein kinase 2), SNCA (α-synuclein), COMT (catechol-O-methyltransferase), BDNF (brain-derived neurotrophic factor), and various genes involved in inflammation/immune responses, such as IL10 (associated with lower risk of cognitive impairment), IL17A (associated with higher risk of cognitive impairment), IL18, and IFNG [1]. While the genetics associated with PD and PD cognitive impairment do not currently have treatment implications, precision medicine approaches applied to PD suggest that targeting specific genetics like GBA hold promise for future treatments [18].

## 2. Current Treatment and Therapies

There are no current disease-modifying treatments for PD-MCI or PDD. There are no medications approved by the U.S. Food and Drugs Administration (FDA) or European Medicines Agency (EMA) for PD-MCI. Rivastigmine is the only FDA- and EMA-approved medication for PDD (Figure 1). Current treatments for PD-related cognitive impairment primarily focus on symptom management, such as treating mood disorders, behavioral disturbance, sleep disorder, and lifestyle modification to improve quality of life (e.g., physical activities, healthy diet and social engagement). While PD is more common in men than women, there currently exist no differences in treatment approaches in men and women with PD. Table 2 summarizes current pharmacologic treatments based on clinical evidence.

Treating cognitive changes in PD starts with an assessment for potentially reversible contributors to cognitive changes. Major pharmacologic culprits are anticholinergic medications (e.g., antihistamines, antispasmodics), benzodiazepines, and opioids (Table 3). Medications used to treat motor symptoms in PD can also contribute to cognitive symptoms. In the context of cognitive complaints, the general order in which PD medications are discontinued is as follows: anticholinergics, MAO-B inhibitors, amantadine, dopamine agonists, and COMT inhibitors, followed by reduced levodopa [19]. Laboratory assessment for contributors to cognitive impairment commonly includes a complete blood count, serum electrolytes, glucose, blood urea nitrogen/creatinine, folate, B12, and thyroid function testing [20]. Addressing depression and hearing loss are also part of a comprehensive approach to addressing cognitive impairment.

### 2.1. Studied Treatments for Parkinson’s Disease—Mild Cognitive Impairment

#### 2.1.1. Pharmacologic Treatments for Parkinson’s Disease—Mild Cognitive Impairment

Research investigated the effects of four medications and two supplements for the treatment of PD-MCI, but no agent has consistently positive results. The transdermal (patch) form of rivastigmine, a cholinesterase inhibitor (4.6–9.5 mg/24h), had no statistically significant benefits in a 24-week randomized controlled trial (RCT) enrolling 28 individuals with PD-MCI [21]. The endpoints included the Alzheimer’s Disease Assessment Scale-clinicians’ global impression of change scales (ADCS-CGIC), the MoCA, NeuroTrax computerized cognitive testing, Dementia Rating Scale-2 (DRS-2), and the Gordon Diagnostics System. An initial RCT (*n* = 48) of rasagiline (1 mg daily), a monoamine oxidase B inhibitor, demonstrated significant improvement in digit span and verbal fluency tasks at 12 weeks in the rasagiline group [22]. However, a subsequent RCT (*n* = 151) showed no significant between-group differences in Change in Scales for Outcomes of Parkinson’s Disease-Cognition, MoCA, Penn Daily Activities Questionnaire, or ADCS-CGIC (MCI) scores at 24 weeks [23]. Clinicaltrials.gov lists a third study aiming to evaluate rasagiline’s effect on cognition in PD-MCI (NCT01497652, *n* = 34), but the study finished in 2015 and no results were published.

Researchers assessed atomoxetine, a selective norepinephrine reuptake inhibitor commonly used in attention deficit hyperactive disorder, in multiple studies of individuals with PD. In a small (*n* = 12) open-label, uncontrolled, flexible dose study involving individuals with PD and executive dysfunction without dementia (MMSE ≤ 25), atomoxetine (25–100 mg/day) improved executive function on the CGIC scale, Frontal Systems Behavior Scale (FrSBE), and Connors Adult Attention-Deficit/Hyperactivity Disorder Rating Scale (CAARS), a subjective measure of impulsivity and attention, at 8 weeks [24]. An RCT involving individuals with PD-MCI (*n* = 30), atomoxetine (80 mg/day) showed no difference between the treatment and control groups in the primary cognitive outcome, a composite score based on a battery of standardized executive function tests, but the atomoxetine group had significant improvement on CAARS [25]. In an RCT (*n* = 55) of atomoxetine for the treatment of depression in individuals with PD and a MMSE >14, those who received atomoxetine (target dose 80 mg/day) had a greater mean change in MMSE at 8 weeks compared to the placebo group (1.31, *p* = 0.003) [26]. A meta-analysis of these three studies concluded that atomoxetine improved global cognition (statistically significant large effect size), but changes in other categories (e.g., complex attention, executive function) were non-significant [27]. An RCT (*n* = 30) evaluating atomoxetine’s effect on cognition in PD-MCI was completed but no results were published (NCT01738191; last accessed on 17 January 2021).

Creatine and coenzyme Q10 are important substrates involved in energy conversion in the mitochondria and have an indirect antioxidant effect. A Chinese study (*n* = 75) found that creatine (5 g twice a day) and coenzyme Q10 (100 mg three times per day) improved MoCA scores in PD-MCI after 12 and 18 months [28]. The study did not adjust for baseline MoCA performance, however, and larger scale follow up studies have not been done to date.

#### 2.1.2. Non-Pharmacologic Treatments for Parkinson’s Disease—Mild Cognitive Impairment

##### Cognitive Interventions

Cognitive interventions are divided into three types: cognitive stimulation, cognitive training, and cognitive rehabilitation [29]. Cognitive stimulation consists of non-specific stimulation of cognitive and social functioning. Cognitive training uses standardized cognitive tasks on the computer or on paper. Cognitive rehabilitation targets specific areas of difficulty in activities of daily living to improve function. A recent Cochrane systemic review evaluated seven RCTs on cognitive training for PD-MCI and PDD [29]. Four of the studies targeted persons with PD-MCI, two targeted individuals with PD without dementia (PD-MCI + PD-normal cognition), and one study included persons with PDD. Interventions varied from tailored exercises that stimulated executive function, attention, shifting ability, visuospatial function, multi-domain online computer-based training over 4–8 weeks. The review concluded that there was no significant improvement in global cognition with the investigated approaches. One RCT in the systematic review (*n* = 70) investigated a structured cognitive training program NEUROvitalis that focused on the cognitive domains frequently impaired in PD (attention, memory, executive functions). The study compared the treatment to unstructured training (“Mentally fit,” no specific targeted cognitive domains) and to a non-intervention control group. NEUROvitalis improved short-term memory and working memory in individuals with PD without dementia (MMSE ≥25), compared to the control group after 6 weeks (*p* < 0.01 and *p* < 0.05, respectively) [30]. Forty-seven individuals from the same study were reexamined at 1 year, and both the structured (NEUROvitalis) and non-structured (Mentally fit) cognitive training programs for 6 weeks improved overall cognitive function (MMSE and DemTect). The MCI risk was higher in the control group versus treatment groups (40.0%, versus 18.2% in NEUROvitalis and Mentally Fit, respectively) [31]. Other studies are underway, including an RCT that evaluates the effect of cognitive rehabilitation compared to standard supportive care (NCT03335150) and an RCT for targeted intervention strategy on prospective memory (NCT01469741) in PD-MCI and PD-normal cognition. Other studies in persons with PD-MCI without published data include the following: computer-based cognitive training (NCT02225314), neurocognitive and supportive therapy (NCT01646333), cognitive behavioral therapy (NCT02048605), and cognitive training (NCT04474379, NCT03582670).

##### Exercise and Physical Therapy

Different types of physical exercises have been evaluated for their effects on cognition, including treadmill training, dance, stationary bicycle, Wii Fit, and Tai chi. A systematic review of RCTs on the effects of physical exercise on cognition in PD (those with normal cognition and PD-MCI) showed that physical exercise improved global cognition, processing speed, attention and mental flexibility. Treadmill use 3 times per week for 60 min improved cognition with the most significance [32]. In persons with mild to moderate PD and a MMSE >24, treadmill walking (45 min, 3 days per week for 3 weeks) significantly improved executive function (measured by the Frontal Assessment Battery, trail-making and memory interference test) [33]. However, MoCA scores did not differ between the intervention and control groups. One RCT (*n* = 76) evaluated the effects of aerobic exercise (recumbent bicycle) and goal-based exercises (walking, muscle toning, and stretching exercises) for 60 min per session, 3 times per week for 12 weeks, compared to age-matched controls in individuals with PD with normal cognition and those with PD-MCI [34]. In both the cognitively normal and MCI groups, aerobic exercise improved executive function (Stroop test) compared to the goal-based group, but not better than the control group. In persons with PD-MCI, the aerobic group outperformed the control and goal-based groups on the trail-making test (a test of executive function and processing speed). There were no differences between groups in memory and language. A meta-analysis in **Tai chi and Qigong** exercises in PD, including two studies that assessed cognition, found that they improved most motor function, depression and quality of life, but not cognition [35]. Notably, study populations were heterogeneous, including persons with PD with normal cognition and PD-MCI. Two RCTs assessed Argentine **tango** dancing on cognition. One study (*n* = 33) showed that tango (90 min per session, 20 sessions over 12 weeks) improved spatial cognition (measured by Brook’s spatial task) in persons with PD without dementia (baseline MoCA > 24), but there was no difference in global cognition (measured by MoCA) between the treatment and control groups [36]. Another RCT (*n* = 33) found that tango (60 min per session, twice a week for 12 weeks) did not improve global cognition (measured by MoCA) in persons with PD without dementia compared to the home exercise control group [37]. A current RCT is investigating the effect of Rock Steady Boxing (versus sensory attention focused exercise) on motor and cognitive outcome in persons with idiopathic PD (NCT03618901).

An RCT (*n* = 40) found that intensive physical therapy (60 min per session, 6 times per week for 4 weeks, including aerobic exercises, treadmill training and exercise intervention program) improved global cognition (measured by MoCA) and attention/working memory (verbal and digit span) in persons with PD-MCI [38]. Other clinical trials are investigating the effects of certain types of exercise on cognition in persons with PD-MCI (NCT02267785) and some are utilizing home-based physical therapy (NCT02248649). A clinical trial is evaluating the roles of the supplement carnosine and physical exercise on working memory and motor functions in persons with MCI, early-stage PD, and subjective cognitive impairment (NCT03330470).

In summary, physical exercise shows some benefit on cognition in individuals with PD and normal cognition and those with PD-MCI, but the effects vary by exercise types. Aerobic exercises such as treadmill, recumbent bicycle, and tango most consistently improve executive function in the identified studies, but the effects on global cognition and other cognitive domains vary. Intensive physical therapy may improve global cognition, attention/working memory in persons with PD-MCI.

##### Device-Related Interventions

Transcranial magnetic stimulation (TMS) has been studied in PD for the treatment of motor, mood and cognitive symptoms. There is no definitive evidence for repetitive TMS (rTMS) in improving cognition in PD-associated cognitive impairment. One study showed that repeated intermittent “theta burst” stimulation of the left dorsolateral prefrontal cortex (DLPFC) improved cognition and visuospatial function lasting up to 1 month in persons with PD-MCI [39]. However, another study (*n* = 46) found that stimulation of the bilateral DLPFC did not improve overall cognition (primary outcome: DRS-2) or Clinical Global Impression of Improvement in persons with PD-MCI compared to the control group [40]. Clinical trials are ongoing to assess rTMS stimulation of bilateral (NCT02346708) and left DLPFC in PD-MCI (NCT03836950).

Transcranial direct current stimulation (tDCS) over the prefrontal cortex improved executive function, as measured in trail-making tasks, but not others (e.g., Stroop test, Wisconsin Card Sorting Test) in individuals with PD without dementia [41]. An RCT (*n* = 22) showed that the combination of cognitive training and transcranial direct current stimulation (tDCS) to the left DLPFC in individuals with PD-MCI (5 days/week for 2 weeks) improved phonemic verbal fluency compared to cognitive training alone, and this effect persisted at 3 months. However, there were no between-group differences in other primary outcome measures in language, attention, and executive functions [42]. Another RCT (*n* = 24) in individuals with PD-MCI showed that the combination of tDCS over the left DLPFC and cognitive training for 4 weeks did not improve memory and cognition at 16 weeks [43]. One RCT (*n* = 42) showed that cognitive training (45 min, 3 times per week for 4 weeks) and tDCS of the left DLPFC (20 min, once per week for 4 weeks) in isolation or combined both improved various cognitive and functional outcomes at 5 and 12 weeks, and combined interventions had greater effects [44]. The standard cognitive training group improved on memory, ADLs and quality of life. The tailored cognitive training group improved on attention/working memory and quality of life. The tDCS group improved on attention/working memory and memory. The standard cognitive training + tDCS group improved on executive function, attention/working memory and ADLs. The tailored cognitive training + tDCS group improved on executive function, attention/working memory and memory. Other clinical trials are ongoing to evaluate the effect of tDCS on cognition in PD-MCI (NCT03191916, NCT04171804) and individuals with PD without dementia (NCT03025334). Combination therapies in persons with PD-MCI, such as cognitive training with tDCS, have been heterogeneous and have shown mixed results [42,43,44]. An RCT (*n* = 20) showed that tDCS over the medial frontal cortex improved the Theory of Mind, the ability to understand and predict others’ behaviors in individuals with PD-MCI [45]. Pathologic oscillatory activities exist in the thalamo-cortical region in PD, thus transcranial alternating current stimulation (tACS) was investigated for potential disease modulation. An RCT (*n* = 15) using personalized transcranial alternating current stimulation, or tACS (location of stimulation was individually defined based on EEG data) 5 days/week for 2 weeks, did not show significant cognitive benefit in persons with PD with MMSE ≥ 23 [46].

Ongoing research is evaluating the role of vestibular nerve stimulation (NCT04450550), transcranial ultrasound (NCT04250376), and deep brain stimulation (DBS) of the globus pallidus interna (GPi) and the nucleus basalis of Meynert (NBM) (NCT04571112) or NBM alone (NCT02924194) in individuals with PD without dementia.

#### 2.1.3. Clinical Treatment Approaches for PD-MCI

The 2019 Movement Disorders Society update on evidence-based treatments for non-motor symptoms of PD concluded that there is currently insufficient evidence to support any pharmacologic (rivastigmine, rasagiline) or non-pharmacologic (transcranial direct-current stimulation, cognitive rehabilitation) treatment for non-dementia cognitive impairment in PD. All reviewed approaches were labeled investigational [47]. However, research suggests that there may be a role for physical exercise and therapy in targeting cognitive outcomes in populations with PD without dementia [32]. Aerobic exercise (treadmill and recumbent bicycle) improved executive function in PD-MCI, while intensive physical therapy improved global cognition. More research is needed to identify pharmacologic and non-pharmacologic strategies for targeting PD-MCI, as well as combination approaches. In the meantime, particularly given the increasing research showing the benefits of exercise on PD more generally, it is reasonable for clinicians to recommend exercise as a strategy to target early cognitive changes in PD.

### 2.2. Studied Treatments for Parkinson’s Disease Dementia

#### 2.2.1. Pharmacologic Treatments for Parkinson’s Disease Dementia

Cholinesterase inhibitors are the most frequently studied medications for PDD. Rivastigmine is the only cholinesterase inhibitor that is approved by the FDA for use in PDD. It is also approved for such use by the EMA, Canada and Japan. In an RCT of rivastigmine (3–12 mg) in individuals with mild-to-moderate PDD (*n* = 410), rivastigmine improved cognition (measured by ADAS-cog, MMSE, and Delis–Kaplan Executive Function System (D-KEFS) verbal fluency and ten-point clock drawing) as well as global function (ADCS-CGIC and ADCS-ADL) [48]. In a 76-week, open-label randomized study of rivastigmine capsule and patch formulations in individuals with mild-to-moderate PDD (*n* = 583), individuals receiving rivastigmine capsules had significantly better Mattis Dementia Rating Scale scores (weeks 24–76) and better ADCS-ADL scores at weeks 52 and 76 [49]. There were no differences in efficacy between the capsule and patch in persons with MMSE > 21, but individuals with lower MMSE scores benefited more from the capsules [49].

Donepezil showed conflicting results in PDD studies. An RCT (*n* = 16) found that donepezil 2.5–10 mg/day improved memory in persons with PD-MCI and PDD on the Dementia Rating Scale (DRS) but resulted in no significant changes in global cognitive status, attention, executive function, memory or visuospatial functions [50]. Another RCT (*n* = 22) showed that donepezil 2 to 10 mg/day for 10 weeks improved performance on the MMSE but not on the Mattis Dementia Rating Scale or ADAS-cog [51]. The EDON study (*n* = 550) found that donepezil 5 or 10 mg for 24 weeks did not improve its primary endpoints, ADAS-cog and Clinician’s Interview-based Impression of Change plus caregiver input (CIBIC+). After removing the treatment-by-country interaction from the model, 10 mg donepezil had a dose-dependent benefit in the CIBIC+ scores. Both 5 mg and 10 mg doses improved secondary endpoints: MMSE, D-KEFS, and Brief Test of Attention [52]. Galantamine (8 mg twice daily) was assessed in a single-open label study (*n* = 41). Participants reported improvements on the MMSE, ADAS-cog, clock drawing test, and Frontal Assessment Battery [53].

Memantine is an NMDA receptor antagonist considered investigational for use in PDD [47]. A metanalysis included three studies: one RCT specifically for PDD, and two for mixed dementia with Lewy bodies (DLB) and PDD [54]. There were no significant differences in the ADCS-CGIC scores in the memantine-treated group (20 mg/day) versus the control group for up to 24 weeks. One study showed improved attention and episodic memory in PDD [55].

#### 2.2.2. Non-Pharmacologic Treatments for Parkinson’ Disease Dementia

Studies on the effects of non-pharmacologic approaches on cognition in individuals with PDD are scarce. One systemic review that examined the effect of exercise in Lewy body dementia (LBD) included five studies and only three included individuals with PDD (total of 10 individuals with PDD) [56].

Deep brain stimulation (DBS): An RCT (*n* = 6) evaluating bilateral NBM DBS in persons with PDD found no differences in the primary cognitive outcomes (California Verbal Learning Test, WAIS-III digit span, verbal fluency, Posner covert attention test, and simple and choice reaction times) between the treatment and control groups after 6 weeks [57]. Other RCTs are ongoing to evaluate subthalamic nucleus (STN) and NBM stimulation in treating motor and cognitive symptoms in PDD (NCT02589925). Other non-pharmacologic treatments being studied for PDD include transcranial ultrasound (NCT04250376) and temporary blood brain barrier (BBB) disruption using non-invasive ultrasound (NCT03608553).

#### 2.2.3. Clinical Treatment Approaches for Parkinson’s Disease Dementia

The 2019 Movement Disorders Society update on evidence-based treatments for non-motor symptoms of PD concluded that there is only sufficient evidence to support rivastigmine as efficacious for PD dementia. There is insufficient evidence of the efficacy of donepezil, galantamine, or memantine. The update identified rivastigmine as clinically useful. The other cholinesterase inhibitors were classified as possibly useful, given their antidementia benefit outside PD. Memantine was labeled as investigational [47]. Research on non-pharmacologic approaches for PDD are lacking and there is insufficient evidence to support the use of non-pharmacological procedures (DBS, tACS) for PDD.

### 2.3. Treatments for Non-Cognitive Symptoms

A plethora of non-cognitive symptoms can manifest in PD-MCI and PDD, including fatigue, depression, anxiety, apathy, psychosis, REM sleep behavioral disorder, insomnia, and impulse control disorder. Treatments for these symptoms are a key part of management of cognitive impairment in PD but are beyond the scope of this review. Clinicians should screen for non-motor symptoms accompanying cognitive impairment in PD and treat them with pharmacologic and non-pharmacologic strategies, as needed.

## 3. Developing Research

In addition to aforementioned therapies in PD-MCI and PDD, studies are investigating treatments for cognitive impairment and dementia related to PD using various approaches, including (1) drug repurposing, (2) novel medications and (3) non-pharmacologic interventions. Clinical trials in pharmacologic and non-pharmacologic treatments are summarized in Table 4 and Table 5, respectively.

### 3.1. Drug Repurposing

Researchers are repurposing multiple pharmacologic agents for potential use in PD cognitive impairment. Ambroxol is an over-the-counter expectorant that is a pharmacological chaperone of B-glucocerebrosidase (GCase), coded by the gene GBA1. GBA1 mutations are a strong risk factor for PDD [58]. Ambroxol showed promise in mouse models and is being studied in a phase 2 RCT for persons with mild to moderate PDD (NCT02914366) [58]. Ceftriaxone, a third-generation cephalosporin antibiotic, is being studied in a phase 2 RCT in PDD given its potential neuroprotective effect by reducing glutamatergic hyperactivity and excitotoxicity (NCT03413384). Nilotinib is a tyrosine kinase Abelson inhibitor used to treat chronic myeloid leukemia, but it also increases dopamine levels. After a small proof-of-concept trial (*n* = 12) [59], a phase 2 RCT in persons with PD-MCI (MoCA score ≥ 22) found that nilotinib did not improve MoCA scores in the 150 mg group and significantly worsened in the 300 mg treatment group [60]. Intranasal insulin was previously shown to improve visuospatial and verbal working memory in persons with MCI and Alzheimer’s disease [61,62]. A proof-of-concept RCT (*n* = 16) in individuals with PD without dementia found that daily intranasal insulin (40 international units) improved letter fluency and Hoehn and Yahr scores but not MoCA scores [63]. No other neuropsychological assessment was performed outside of letter fluency.

One study evaluated the effects of atomoxetine (10–30 mg twice a day) and rivastigmine (1.5–4.5 mg twice a day) on attention in persons with PD without dementia (NCT01340885, *n* = 9). Results are not yet published.

### 3.2. Novel Medications

SYN120, a dual serotonin 5-HT6/5-HT2A antagonist, did not improve cognition (measured by Computerized Drug Research Cognition Battery, CDR episodic memory and ADAS-Cog) in persons with PDD compared to the placebo after 16 weeks in a proof-of-concept RCT (SYNAPSE trial; NCT02258152; *n* = 82) [64]. ANAVEX2-73 (Blarcamesine), an agonist of the intracellular sigma-1 chaperone protein, is in a phase 2 RCT in individuals with PDD (NCT03774459) as well as an open-label extension to evaluate for its safety and efficacy of daily treatment (NCT04575259). LY3154207 (mevidalen), a dopamine receptor D1 enhancer, is in a phase 2 trial to evaluate its effect on cognition in mild-to-moderate dementia due to LBD associated with idiopathic PD and DLB (NCT03305809). RO7046015/PRX002 (Prasinezumab) is an anti-α-synuclein monoclonal antibody (mAb) whose precursor reduced α-synuclein accumulation and improved cognition in animal models of PD [65]. Studies showed that intravenous PRX002 was safe in healthy adults [65] and in persons with mild to moderate idiopathic PD [66]. A trial is investigating its efficacy in early PD (primary outcome: change in UPDRS; secondary outcome includes change in MoCA score; NCT03100149). ENT-01 is in a phase 1 RCT to assess its effect on cognition in PDD (NCT03938922). GRF6021 is a plasma derivative being investigated for its safety and tolerability in persons with PD-MCI in a phase 2 RCT (NCT03713957). NMDA receptor modulators, NYX-458 and D-amino acid oxidase (DAAO) inhibitor (DAAOI-P), are under investigation for PD-MCI (NCT04148391) and PDD (NCT04470037).

### 3.3. Non-Pharmacologic Interventions

Active clinical trials for non-pharmacologic interventions for PD-MCI and PDD are reviewed in Section 2.1.2 and Section 2.2.2, respectively.

### 3.4. Challenges to Study Design in PD Cognitive Impairment

There are numerous challenges to clinical trials for individuals with PD and cognitive impairment. First, robust understanding of pathophysiologic mechanisms to guide intervention development is lacking. It is also possible that effective therapy will need to employ more than one type of treatment, such as combined pharmacologic cognitive interventions and exercise. From the perspective of study design, current studies sometimes mix populations (e.g., PD-MCI plus PD with normal cognition). It remains uncertain how best to identify populations for particular studies, such as whether to study PD without dementia (normal cognition and PD-MCI), PD with cognitive impairment (PD-MCI plus PDD), or the conditions separately. There are also debates regarding whether individuals with PDD should be combined with individuals with dementia with Lewy bodies for trials, as they both fall under the umbrella of Lewy body dementia [67]. Cognitive impairment in PD is a spectrum and how PD-MCI and PDD are operationalized for study design may affect results. Other aspects of PD cognitive impairment need also to be considered in trial design, such as whether researchers need to account for the presence or absence of concomitant AD pathology [67]. Features of PDD, such as fluctuations, can also negatively affect accurate clinical trial measurements, and study participation requirements (e.g., travel, study visit frequency) can affect trial enrollment, retention, and outcomes [67]. Finally, most measures used in studies of cognitive impairment in PD need further research to assess the validity, reliability, and clinically meaningful change of these measures when used in populations with PD and cognitive impairment [68].

## 4. Conclusions and Future Directions

Despite tremendous research efforts focused on PD-related cognitive impairment, current options are limited. When treating individuals with PD-MCI, clinicians should counsel patients regarding the absence of established pharmacologic therapies. Emerging research regarding the potential benefits of exercise in PD-MCI—in addition to research showing the benefits of exercise in PD more generally—suggest that clinicians should recommend exercise, especially the aerobic type, for individuals with PD and cognitive impairment. For individuals with PDD, rivastigmine is the only currently FDA-approved treatment for PDD, though other cholinesterase inhibitors are also sometimes used clinically. Memantine is labeled as investigational by the *Movement Disorders Society Evidence Review*, but is sometimes also tried in clinical practice. Current research approaches include both drug repurposing and identifying drugs with novel mechanisms of action. Additional non-pharmacological intervention studies are also needed, especially for PD dementia, where such research is particularly lacking.

## Figures and Tables

**Figure 1 behavsci-11-00054-f001:**
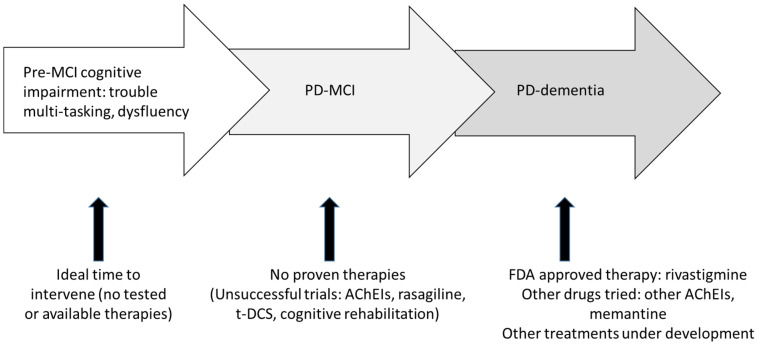
Approach to treating cognitive impairment in Parkinson’s disease. AchEI: acetylcholinesterase inhibitors. T-DCS: transcranial direct current stimulation.

**Table 1 behavsci-11-00054-t001:** Neurotransmitters targeted for pharmacologic treatments for Parkinson’s disease cognitive changes.

Target Neurotransmitter	Drugs	Status	
		PD-MCI	PDD
Acetylcholine	Rivastigmine	Experimental	FDA-approved
	Donepezil		Off-label
	Galantamine		Off-label
NMDA	Memantine		Experimental
	NYX-458	Experimental	
	D-amino acid oxidase inhibitor (DAAOI-P)		Experimental
Dopamine	MAO-B inhibitors (e.g., rasagiline)	Investigational	
	Mevidalen (LY3154207) (D1 receptor positive allosteric modulator)		Experimental
Serotonin	SYN120 (dual 5-HT6/5-HT2A antagonist)		Experimental
Norepinephrine		Experimental	

MAO: monoamine oxidase, NMDA: N-methyl-d-aspartate.

**Table 2 behavsci-11-00054-t002:** Clinically available pharmacologic treatments for cognition in Parkinson’s disease mild cognitive impairment and Parkinson’s disease dementia.

Category	Specific Agents	MCI	Dementia	Most Common Adverse Effects	Severe but Rare Adverse Effects
Cholinesterase inhibitors	Rivastigmine	Investigational ^3^	Clinically useful ^1^	Capsules: Nausea, Vomiting, weight loss Patch: nausea, vomiting, falls	Capsules: atrial fibrillation, myocardial infarction, hypokalemia, transient ischemic attack, seizures. Patch: dehydration
	Donepezil	Not studied	Possibly useful ^2^	Nausea, diarrhea, vomiting	Gastrointestinal hemorrhage, heart block, torsades de pointes
	Galantamine	Not studied	Possibly useful ^2^	Nausea, vomiting, diarrhea	Syncope, Stevens–Johnson syndrome, gastrointestinal hemorrhage, seizure
NMDA Receptor Antagonist	Memantine	Not studied	Investigational ^3^	Diarrhea, constipation, confusion, dizziness	Stroke, seizure, renal failure
Dopaminergic therapy	Rasagiline (monoamine oxidase B inhibitor)	Investigational ^3^	Not studied	Orthostatic hypotension, headache, nausea	Serotonin syndrome
Selective norepinephrine reuptake inhibitor	Atomoxetine	Investigational	Not studied	Increased heart rate, nausea, decreased appetite, xerostomia	Sudden cardiac death, stroke

1: Clinically useful as determined by the *International Parkinson and Movement Disorder Society Evidence-Based Medicine Review*. 2: Possibly useful as determined by the *International Parkinson and Movement Disorder Society Evidence-Based Medicine Review*. 3: Investigational as determined by the International Parkinson and Movement Disorder Society Evidence Based Medicine Review. Inclusion in the table indicates that the pharmacologic agent is clinically available but does not imply regulatory approval for any specific indication. Adverse effects obtained from Micromedex, last accessed on 17 January 2021.

**Table 3 behavsci-11-00054-t003:** Common medications associated with adverse cognitive effects.

Drug Class	Examples
**Anticholinergics**	
Tricyclic antidepressants	Amitriptyline, nortriptyline
First generation antihistamines	Diphenhydramine, hydroxyzine
Bladder antimuscarinics	Oxybutynin, trospium
Antipsychotics	Fluphenazine
Antimuscarinic spasmolytic	Atropine, hyoscyamine
Antiemetics	Meclizine
Muscle relaxants	Tizanidine
Anti-Parkinson	Benztropine, trihexyphenidyl
**Benzodiazepines**	Alprazolam, clonazepam, diazepam, lorazepam
**Opioids**	Codeine, hydrocodone, morphine, oxycodone, tramadol, methadone, fentanyl

This table is non-exhaustive.

**Table 4 behavsci-11-00054-t004:** Clinical trials on pharmacologic treatments for cognitive impairment in Parkinson’s disease.

Condition	Study Title	Drug Name	Trial Number	Start Date	Status/Phase	Location	Last Accessed
Investigation of Current drugs							
PD-MCI	A Double-Blind Placebo Controlled Trial Evaluating Rasagiline Effects on Cognition in Parkinson’s Disease Patients with Mild Cognitive Impairment Receiving Dopaminergic Therapy (RECOGNIZE)	Rasagiline	NCT01497652	January 2012	Completed/4	USA	17 January 2021
PDD	Memantine for Treatment of Cognitive Impairment in Patients with Parkinson’s Disease and Dementia	Memantine	NCT00294554	September 2008	Completed/N/A	USA	17 January 2021
Drug Repurposing							
PDD	Ambroxol as a Treatment for Parkinson’s Disease Dementia	Ambroxol	NCT02914366	November 2015	Recruiting/2	Canada	17 January 2021
PDD	To Assess the Efficacy and Safety of Ceftriaxone in Patients with Mild to Moderate Parkinson’s Disease Dementia	Ceftriaxone	NCT03413384	15 February 2019	Recruiting/2	Taiwan	17 January 2021
PD without dementia	Cognitive Dysfunction in PD: Pathophysiology and Potential Treatments, a Pilot Study	Atomoxetine, rivastigmine	NCT01340885	2011	Completed/4	USA	17 January 2021
Novel Medications							
PDD	SYN120 Study to Evaluate Its Safety, Tolerability and Efficacy in Parkinson’s Disease Dementia (SYNAPSE)	SYN120	NCT02258152	22 December 2014	Completed/2	USA	17 January 2021
PDD	ANAVEX2-73 Study in Parkinson’s Disease Dementia	ANAVEX2-73	NCT03774459	9 July 2018	Completed/2	Multiple, international	17 January 2021
PDD	OLE Study for Patients with Parkinson’s Disease with Dementia Enrolled in Study ANAVEX2-73-PDD-001	ANAVEX2-73	NCT04575259	10 October 2019	Recruiting/2	USA	17 January 2021
PDD	Effect of LY3154207 on Cognition in Mild-to-Moderate Dementia Due to Lewy Body Dementia (LBD) Associated with Idiopathic Parkinson’s Disease (PD) or Dementia with Lewy Bodies (DLB)	LY3154207	NCT03305809	November 2017	Completed/2	USA, Canada, Puerto Rico	17 January 2021
PD ^1^	A Study to Evaluate the Efficacy of Prasinezumab (RO7046015/PRX002) in Participants with Early Parkinson’s Disease (PASADENA)	Prasinezumab	NCT03100149	June 2017	Active, not recruiting/2	Multiple, international	18 January 2021
PDD	A Multicenter, Open Label Study to Evaluate Tolerability and Efficacy of Orally Administered ENT-01 for the Treatment of Parkinson’s Disease Dementia.	ENT-01	NCT03938922	13 June 2019	Active, not recruiting/1	USA	18 January 2021
PD-MCI and probable or possible PDD	A Study to Assess the Safety of GRF6021 Infusions in Subjects with Parkinson’s Disease and Cognitive Impairment	GRF6021	NCT03713957	November 2018	Completed/2	Multiple, international	18 January 2021
PD-MCI	A Study to Evaluate NYX-458 in Subjects with Mild Cognitive Impairment Associated with Parkinson’s Disease	NYX-458	NCT04148391	November 2019	Active, not recruiting/2	USA	18 January 2021
PDD	Multidisciplinary Study of Novel NMDA Modulation for Neurodegenerative Disorder	DAAOI-P	NCT04470037	April 2016	Recruiting/2	Taiwan	18 January 2021

^1^: studies in individuals with Parkinson’s disease without specifying the presence of MCI or dementia.

**Table 5 behavsci-11-00054-t005:** Clinical trials on non-pharmacologic treatments for cognitive impairment in Parkinson’s disease.

Condition	Study Title	Therapy	Trial Number	Start Date	Status/Phase	Location	Last Accessed
PD-MCI	Cognitive Rehabilitation for Individuals with Parkinson’s Disease and MCI	Cognitive training	NCT03335150	November 2015	Active, not recruiting/N/A	USA	18 January 2021
PD with normal cognition, PD-MCI	Rehabilitation of Everyday Memory Impairment in Parkinson’s Disease: A Pilot Study	Cognitive training	NCT01469741	2011	Completed/N/A	USA	18 January 2021
PD-MCI	Computer-based Cognitive Training for Individuals with Parkinson’s Disease	Computer-based cognitive training	NCT02225314	2012	Completed/N/A	USA	18 January 2021
PD-MCI	A Trial of Neurocognitive and Supportive Therapy Interventions for Individuals with Parkinson’s Disease	Cognitive rehabilitation	NCT01646333	July 2012	Completed/N/A	USA	18 January 2021
PD-MCI, mild PDD	Training of Psychosocial Skills Based on Cognitive Behavioral Therapy for Patients with Parkinson’s Disease (CBT)	Cognitive Behavioral Therapy	NCT02048605	February 2014	Active, not recruiting/N/A	Switzerland	18 January 2021
PD with normal cognition, PD-MCI	Prospective Memory Impairment in Parkinson’s Disease-related Cognitive Decline: Intervention and Mechanisms	Cognitive training	NCT04474379	January 2021	Not yet recruiting/N/A	USA	18 January 2021
PD-MCI	Prospective Memory Training in Parkinson’s Disease	Cognitive training	NCT03582670	October 2017	Completed/N/S	USA	18 January 2021
PD-MCI	Cognitive Rehabilitation for Veterans with Parkinson’s Disease	Cognitive training	NCT03836963	January 2020	Recruiting/N/A	USA	18 January 2021
PD-MCI	Exercise Targeting Cognitive Impairment in Parkinson’s Disease	Exercise	NCT02267785	October 2014	Active, not recruiting/N/A	USA	18 January 2021
PD-MCI	A Telemedicine Intervention to Improve Cognitive Function in Patients With PD	Exercise	NCT02248649	December 2014	Completed/N/A	USA	18 January 2021
PD, MCI ^1^	Molecular Mediators of Physical Exercise and Carnosine Induced Effects in Patients with Preclinical and Early-Stage Neurodegenerative Disease	Exercise, carnosine	NCT03330470	January 2017	Unknown/N/A	Slovakia, Taiwan	18 January 2021
PDD ^2^	Tailored Music Therapy for Dementia	Music	NCT03011723	January 2017	Active, not recruiting/N/A	Norway	18 January 2021
PD ^3^	Double-blind, Randomized Controlled Trial to Demonstrate Efficacy of Celeste^®^ Specialized Phototherapy in Treating Parkinson’s Disease.	Phototherapy	NCT04453033	November 2020	Not yet recruiting/N/A	N/A	18 January 2021
PDD	Combined Subthalamic and Nucleus Basalis Meynert Deep Brain Stimulation for Parkinson’s Disease with Dementia (Dempark-DBS)	DBS	NCT02589925	October 2016	Active, not recruiting/N/A	Germany	18 January 2021
PD-MCI	GPi+NBM DBS in Parkinson’s Disease with Mild Cognitive Impairment (2T-DBS)	DBS	NCT04571112	December 2017	Active, not recruiting/N/A	Canada	18 January 2021
PD-MCI	Deep Brain Stimulation of the NBM to Treat Mild Cognitive Impairment in Parkinson’s Disease	DBS	NCT02924194	September 2016	Recruiting/N/A	USA	18 January 2021
PD-MCI	Cortical Physiology as a Therapeutic Target in Parkinson’s Disease Related Dementia and Cognitive Dysfunction	TMS	NCT02346708	December 2018	Active, not recruiting/N/A	USA	18 January 2021
PD-MCI	rTMS as a Cognitive Rehabilitation Approach in Veterans with Parkinson’s Disease	TMS	NCT03836950	April 2020	Recruiting/1 & 2	USA	18 January 2021
PD-MCI	Transcranial Direct Current Stimulation for Cognitive Improvement in Parkinson’s Mild Cognitive Impairment (tDCS)	tDCS	NCT03191916	October 2015	Recruiting/N/A	USA	18 January 2021
PD-MCI	Efficacy of Transcranial Direct Current Stimulation in Parkinson’s Disease MCI (PDMCIStim)	tDCS	NCT04171804	January 2019	Recruiting/N/A	Turkey	18 January 2021
PD with normal cognition, PD-MCI	tDCS on Parkinson’s Disease Cognition (tDCS-PD-fMRI)	tDCS	NCT03025334	March 2017	Recruiting/N/A	Canada	18 January 2021
PD-MCI, PDD	The Use of Transcranial Focused Ultrasound for the Treatment of Neurodegenerative Dementias	Transcranial ultrasound	NCT04250376	November 2017	Enrolling by invitation/N/A	USA	18 January 2021
PDD	A Study to Evaluate the Safety and Feasibility of Temporary Blood Brain Barrier Disruption (BBBD) Using Exablate MR Guided Focused Ultrasound in Patients with Parkinson’s Disease Dementia	Transcranial ultrasound	NCT03608553	November 2018	Recruiting/N/A	Spain	18 January 2021
PD with normal cognition, PD-MCI	Electrical Vestibular Stimulation (VeNS) in the Management of Parkinson’s Disease	Vestibular stimulation	NCT04450550	September 2020	Not yet recruiting/N/A	N/A	18 January 2021

^1^: Recruited subjects include persons with subjective cognitive impairment (SCI), MCI, or early-stage PD (Hoehn-Yarh stage 1–2). ^2^: Recruited subjects include persons with Alzheimer’s dementia, vascular dementia, dementia with Lewy bodies, and PDD. ^3^: Recruited subjects include individuals with Parkinson’s disease, not specifically PD-MCI or PDD.

## Data Availability

The data presented in this study are available in this article.

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
