# Peer review of "Treatment of Parkinson’s Disease with Cognitive Impairment: Current Approaches and Future Directions"

_behavsci, 2021, doi:10.3390/bs11040054_

Round 1
Reviewer 1 Report
The authors review the current and future treatments of patients with Parkinson's disease suffering cognitive impairments. Although the underlying mechanisms of cognitive impairment in PD are not entirely understood, there are some shreds of evidence suggesting multifactorial alterations, including the neurophysiological connectivity of brain pathways simulating a prion disease, exosome's dysfunctions, and protein deregulation of the redox homeostasis. Can the authors include a brief paragraph commenting on these issues?
As the authors highlight, there are no current disease-modifying therapies for PD-MCI and PDD. Since the redox homeostasis disturbance has been implicated in Parkinson's disease initiation and progression, is there any promising antioxidant treatment able to modify the disease progression? Can some of the future therapeutic approaches be beneficial also for Alzheimer's disease?
Could the author include a brief paragraph about the role of gut microbiota in Parkinson's disease as well as some proposed therapeutic approaches?
Author Response
The authors review the current and future treatments of patients with Parkinson's disease suffering cognitive impairments. Although the underlying mechanisms of cognitive impairment in PD are not entirely understood, there are some shreds of evidence suggesting multifactorial alterations, including the neurophysiological connectivity of brain pathways simulating a prion disease, exosome's dysfunctions, and protein deregulation of the redox homeostasis. Can the authors include a brief paragraph commenting on these issues?
Response: We now mention these proposed mechanisms in the list of possible contributors to cognitive impairment (line 108-110).
As the authors highlight, there are no current disease-modifying therapies for PD-MCI and PDD. Since the redox homeostasis disturbance has been implicated in Parkinson's disease initiation and progression, is there any promising antioxidant treatment able to modify the disease progression? Can some of the future therapeutic approaches be beneficial also for Alzheimer's disease?
Response: With regard to commenting on experimental therapies, we chose a priori to focus on approaches targeting cognitive impairment in PD currently published or listed as under active study on clinicaltrials.gov. In response to the reviewer’s comments, we have re-added studies of creatine and coenzyme Q10 that included cognitive outcomes (line 250-254). As the request review was specifically about therapies for cognitive impairment in Parkinson disease, and given the vast amount of information on this topic, commenting on therapeutic approaches for Alzheimer’s disease is outside the scope of the current review.
Could the author include a brief paragraph about the role of gut microbiota in Parkinson's disease as well as some proposed therapeutic approaches?
Response: We now mention this in the section on proposed pathophysiology (line 109 and 125-128). However, we identified no cognitive treatments specifically targeting this mechanism and the scope of the review is for treatment of cognitive aspects of PD rather than PD more generally.
Reviewer 2 Report
I found the article very useful in terms of its practicality. It does not provide new knowledge (in the sense that, for example, meta-analyses do), but it synthesizes very well the results of studies on treatment of Parkinson’s disease with cognitive impairment. Moreover, in one article we can find information on a wide range of treatment methods, which is a definite advantage for the reader. I also believe tabular summaries provided in the article are helpful and informative. I think this is an article worth publishing.
Author Response
Thank you very much!
Reviewer 3 Report
The authors have written a comprehensive review; however, it needs a bit of restructuring. A review should bring in the author’s opinions backed by evidence, which can direct the reader to new research pathways – this is missing especially in context of future directions.
Introduction
Introduction should highlight on the purpose of conducting the review, rather than detailing background information about the disease.
There is lot of repetitive and overlapping information from different sources in paragraph 1. It would be better to make summarised or conclusive sentences and cite the references together.
The paragraphs in the introduction section are not well linked to each other. There is a quick transition from one aspect of disease to another. It would be better to add subheadings or do a bit of restructuring.
Section 2
Almost everywhere in this section results from prior studies have been explained in detail, which is good, but the way they have been written is just a summary. It would be better to compare the study results, provide opinion after what has been reviewed, and then give a conclusion regarding the best treatment option.
Lines 136 to 138: Should be re-written.
Section 2.1: The heading provided is not appropriate; it should not be made a separate section with heading.
Section 2.2. and 2.3: The first line gives a conclusion; this should not be there in the beginning of the section. It can be added at the end of the paragraph. First introduce about the treatments and give concluding statements.
Section 2.2.1 This just sounds like a summary of some studies, put together. A conclusion must be provided.
Section 4 (conclusion and future directions): Must be strengthened.
Author Response
The authors have written a comprehensive review; however, it needs a bit of restructuring. A review should bring in the author’s opinions backed by evidence, which can direct the reader to new research pathways – this is missing especially in context of future directions.
Response: We have restructured in various parts of the manuscript as suggested, including adding more subheadings, writing a conclusion section at the end of the treatment sections, etc. Our view is that reviews should comprehensively present the evidence to allow the reader to come to his or her own conclusions. To this end, we provide information about each study to help inform the strength of the evidence. We also cite the Movement Disorders Society evidence-based practice recommendations, which represent the systematic assessments and views of a large group of experts. In this way, the reader does no rely on the opinions of two authors, but rather has both the underlying evidence and the references to a formal recommendation process. However, we have also tried to make some practical suggestions (in addition to citing MDS recommendations) in the new conclusion sections as part of Section 2. We acknowledge that the manuscript benefited from some additional edits, restructuring, and revision of the future directions section and we thank the reviewer for these helpful suggestions.
Introduction: Introduction should highlight on the purpose of conducting the review, rather than detailing background information about the disease.
Response: We have now rewritten the introduction and include the specific goals of the review: “Parkinson’s disease was historically classified as a movement disorder, but cognitive impairment is prevalent, especially later in the disease process. PD-related cognitive impairment is divided into two categories depending on severity and whether the level of cognitive impairment interferes with activities of daily living (ADLs). In PD mild cognitive impairment (PD-MCI), there is no impairment in ADLs. In PD dementia (PDD), ADLs are affected by cognition. The goal of this narrative review is to describe the therapeutic approaches studied for both PD-MCI and PDD, highlight approaches recommended by the Movement Disorders Society, and provide a brief overview of therapies actively under investigation for treating cognitive symptoms in PD. To provide context for current and experimental approaches, we first briefly review the epidemiology and diagnosis of PD-MCI and PDD and proposed mechanisms for cognitive impairment in PD.” (line 59-61)
There is lot of repetitive and overlapping information from different sources in paragraph 1. It would be better to make summarised or conclusive sentences and cite the references together.
Response: We have rewritten the introduction as above. We have also reorganized this section with subheadings to highlight the key types of information presented. Finally, we have attempted to condense references without losing the specific details/numbers provided.
The paragraphs in the introduction section are not well linked to each other. There is a quick transition from one aspect of disease to another. It would be better to add subheadings or do a bit of restructuring.
Response: We added subheadings to the introduction to delineate each section: epidemiology and clinical significance, diagnosis, risk factors, and pathophysiology.
Section 2: Almost everywhere in this section results from prior studies have been explained in detail, which is good, but the way they have been written is just a summary. It would be better to compare the study results, provide opinion after what has been reviewed, and then give a conclusion regarding the best treatment option.
Response: We agree with Reviewer #3 that the text summarizes the study evidence, consistent with the narrative review approach. There cannot be direct comparisons between studies given different methodologies, study populations, etc. Because the requested review was not intended to be a systematic review/meta-analysis or evidence-based guideline, for practice recommendations we cite the Movement Disorders Society publication/recommendations (e.g. page 11, line 406-411; page 13, line 491-497). It is note-worthy that there are very few recommendations for treatment of cognition in PD because of lacking evidence. We summarize an overall treatment approach in Figure 1. We also have added some comments on practical treatment approaches by adding a new summary section at the end of the PD-MCI and PDD sections.
Lines 136 to 138: Should be re-written.
Response: It is unclear to us what this reviewer is referring to. Line 136 in the submitted manuscript in Word is blank, line 137 is “2. Current Treatment and Therapies”. Line 138 “There are no current disease modifying treatments for PD-MCI and PDD”.
Section 2.1: The heading provided is not appropriate; it should not be made a separate section with heading.
Response: We have deleted the section heading.
Section 2.2. and 2.3: The first line gives a conclusion; this should not be there in the beginning of the section. It can be added at the end of the paragraph. First introduce about the treatments and give concluding statements.
Response: We have restructured the sections by moving the summary statements to the end of the subsections.
Section 2.2.1 This just sounds like a summary of some studies, put together. A conclusion must be provided.
Response: We have added a paragraph at the end of this section to summarize the findings.
Section 4 (conclusion and future directions): Must be strengthened.
Response: We have taken two steps to address this comment and also other reviewer comments. We have added a new section (page 18-19) called “Challenges to Study Design in PD Cognitive Impairment” to discuss the challenges to trials in this population. We have also edited the “conclusions and future directions” section to summarize current recommendations for treatment (to the reviewer’s earlier point) and highlighting the need for more studies in both the pharmacologic and non-pharmacologic space.
Reviewer 4 Report
The authors provide a detailed review of current approaches and future directions in the treatment of PD with cognitive impairment (from MCI to dementia) showing important efforts and knowledge in the field. The contents have been organized in two sections. 1. current treatment and therapies, 2. Developing research. However, the reviewer finds their structural approach disbalanced, with a strong and extensive focus on section 1, for current treatment and therapies, but section 2 that is the one looking to the future offering a nice report on tables 4 and 5 but translated to a short discussion analysis.
Probably the first section (2.) could be divided in three ‘conceptual’ sections, gral, MCI, dementia, to this ‘PD x cognition’ is highlighted and overall the different parts are better balanced with the (3.)
The efforts to gather together all the reports could be bolstered to a higher level if the authors can introduce key aspects that present this topic as the challenge it is, and the knowledge they present here as a chance for understanding and a guidance to consider limitations and opportunities in the immediate and near future.
In the next points, some aspects that this reviewer considers can help to achieve the above-mentioned goals (a challenge, new perspectives) are indicated.
- The title needs to reflect the quality of the review’s contents, the new knowledge that the reader will gain after reading this review. As it stands right now (with my respect), the title looks generalist and classical. Would a medical student / reader realize how challenging is this field, or how the new understanding of PD has evolved in recent years or this new understanding will change the treatment of choice, or improve personalized medicine in patients with PD and cognitive impairment? If one omits the year of submission/publication, this title would apply for any year, stage of clinical and drug discovery for PD, so it is important that the title reflects the new knowledge that the review provides on 2021 and what it is useful for, in a specific manner. Use words that provide the idea that we are in a new understanding, revisiting the treatments, new generation treatments, for the third millennium, personalized medicine, or disease modifying medicine. The effort done worth a title that talks about the challenges, new understanding and how this will change the way to treat this specific group of patients with specific needs.
- The historic perspective of the new understanding PD not only as a motor disorder / neurological disease but potentially affecting cognition must be indicated, so the current perspective makes evident it is an important emerging field (as compared to classical concept based on its motor symptoms) and more endeavours are needed.
- Differential diagnosis should be specified, as well as the state of art of early markers of premorbid and prodromal stages (i.e. sensory, biological, emotional, neuroimage,..) as they play a key role to start interventions. This is mentioned in a small sentence somewhere else.
- Sleep disorder is not mentioned in the introduction despite the implications in QoL of patients and caregivers and one must wait to line 143 to find it for the first time. Since it is well developed later, at least some early mention in the introduction would be needed.
- 86 male sex.
Authors should take into advantage of this sex/gender major issue and introduce and/or discuss its implications in gender medicine. Are the drug pharmacokinetics, pharmacodynamics, side effects, etc the same for males and females with PD disease? To which extent prevention/onset/severity treatments differ?
- 94 The underlying mechanism of cognitive impairment in PD is not well 94 understood, and likely includes multiple factors: abnormal protein deposition, loss of dopaminergic neurons, neurotransmitter deficits, synaptic dysfunction, genetics [1], fatty acid oxidation [10], and perhaps inflammation [11,12].
Except for the specific neuronal loss of DA neurons, all the other targets apply for most neurodegenerative disorders, so this sentence should be more specific. Please, keep in mind that the introduction should also provide the state of art of biological targets that will drive the design, repurposing and choice of treatments, which is the final goal of this review. Another example is:
104 There is also significant overlap between PDD and AD pathology, characterized by low CSF levels of amyloid β-42 (Aβ-42) and cortical deposition of Aβ-42 on neuroimaging [11].
Probably, instead of writing such a statement, it would be better to present the same sentence as a challenge or shortcoming for differential diagnosis and its implications with regards to the design or choice of treatment.
- 106 PD dementia correlates with Alzheimer’s pathology, and individuals with both pathologies have more rapid disease progression (mean survival of 4.5 years, 108 versus 10 years in those without AD pathology).
The sentence is mixing the clinical diagnosis (AD) with the biological conceptual frame referring to BA and tau neuropathologies, which are the hallmarks of AD but not the only ones. Therefore, the statement ‘Alzheimer’s pathology’ includes more neurophysiological derangements than those the authors want to refer.
- The work would benefit to discuss how ‘Research Domain Criteria (RDoC, NIMH) that is how ‘Dysfunction and dysregulation at the genetic, neural, and behavioral levels’ may fit in this clinical scenario.
- Table 1. Do the authors consider that psychosis does not imply cognitive changes, so they can go together in a same description as different nosology entities? Please, reconsider the description.
- Considering that the table is aimed to depict the link between targets and design/use of pharmacological treatments, the descriptive “Medication or medication family targeting” (this implies also posology, etc) should better be referred as “Drugs targeting…”
- Table 1. Since this review is aimed to provide a historic perspective, past, present and future of treatments, I’d suggest to add this perspective into the tables. For instance, table 1, to add an extra column with the year of implementation of the strategy or drug.
- The superfixes a,b,c,d are relevant informations. I’d suggest to highlight them, by including 4 columns, one for each stage, so each drug is placed in one category This will also provide the view of the different stages/purposes of drugs. This would be similar to what is done in Table 2.
- Figure 1 on the approach should go before Table 1.
- Figure 1. ‘medications’ should be better replaced by drugs or treatments (if non-pharmacological ones should also be included)
- Table 2. Clinically available pharmacologic treatments ‘studied’
Not sure if the verb studied is appropriate here. - Treating cognitive changes in PD starts with an assessment for reversible 158 contributors to cognitive changes and addressing potential contributors.
Please, note redundance that can be improved.
- 162 and 163 … ‘psychosis and cognitive changes’ . Please, be aware that psychosis involves changes in cognition, so the sentence should be amended in the way suits better to the authors.
- First generation antihistamines Mind the gap
- 176-184 and also 425 - Non-pharmacological treatments are under-represented (only well addressed for PD-MCI) if we consider what the different international PD societies and family associations say to this respect or they do when organizing an important number of their activities to improve emotionality, cognition and QoL of PD patients and caregivers/families independently of the severity of the stage (i.e. dance therapy, etc).
Please, also revise this statement “Studies on the effect of exercise on cognition in individuals with PDD 426 are scarce. One systemic review examined the effect of exercise in LBD included 5 studies and only 3 included individuals with PDD (total of 10 individuals with PDD) [52].”
- Drug repurposing is a common practice in pharmaceutical companies but scarcely addressed or visualized in other research studies. Since the authors’ efforts to summarize the developing research (or emerging fields) in Tables 4 and 5 are a major contribution of the current review and may be a differentiation of previous reviews on the approaches and directions, I’d suggest to emphasize it in subheading 3. Developing research, and instead highlight the three novel strategies: 3. Drug repurposing, novel medications and non-pharmacologic interventions. In a certain extent, this 3rd part is talking about the immediate future.
- The last sentence of the 4rt part, is clear in the relevance of ‘drug repurposing’ and ‘identifying drugs with novel mechanisms of action’. These could also be interesting for the title
Classical treatments, drug repurposing and new disease-modifying treatments to counteract the challenge of Parkinson’s disease with cognitive impairment.
Author Response
The authors provide a detailed review of current approaches and future directions in the treatment of PD with cognitive impairment (from MCI to dementia) showing important efforts and knowledge in the field. The contents have been organized in two sections. 1. current treatment and therapies, 2. Developing research. However, the reviewer finds their structural approach disbalanced, with a strong and extensive focus on section 1, for current treatment and therapies, but section 2 that is the one looking to the future offering a nice report on tables 4 and 5 but translated to a short discussion analysis.
Probably the first section (2.) could be divided in three ‘conceptual’ sections, gral, MCI, dementia, to this ‘PD x cognition’ is highlighted and overall the different parts are better balanced with the (3.)
Response: This review was written for the audience of clinicians looking to treat individuals with PD and cognitive impairment rather than scientists looking to learn about new treatment pathways. Thus, the focus of the paper is on the active/current treatment approaches and then concluding with a look to the future. Because these approaches may or may not be effective – and because they are not currently approved and thus are of limited active clinical relevance – we maintain the focus on current treatment and therapies. To better explain the focus of the article, we have added a sentence to the introduction regarding the primary purpose of the current review: “The goal of this narrative review is to describe the therapeutic approaches studied for both PD-MCI and PDD, highlight approaches recommended by the Movement Disorders Society, and provide a brief overview of therapies actively under investigation for treating cognitive symptoms in PD.” (line 55-59).
The efforts to gather together all the reports could be bolstered to a higher level if the authors can introduce key aspects that present this topic as the challenge it is, and the knowledge they present here as a chance for understanding and a guidance to consider limitations and opportunities in the immediate and near future.
Response: We have added a completely new section (pages 18-19) called “challenges to study design in PD cognitive impairment.”
In the next points, some aspects that this reviewer considers can help to achieve the above-mentioned goals (a challenge, new perspectives) are indicated.
1. The title needs to reflect the quality of the review’s contents, the new knowledge that the reader will gain after reading this review. As it stands right now (with my respect), the title looks generalist and classical. Would a medical student / reader realize how challenging is this field, or how the new understanding of PD has evolved in recent years or this new understanding will change the treatment of choice, or improve personalized medicine in patients with PD and cognitive impairment? If one omits the year of submission/publication, this title would apply for any year, stage of clinical and drug discovery for PD, so it is important that the title reflects the new knowledge that the review provides on 2021 and what it is useful for, in a specific manner. Use words that provide the idea that we are in a new understanding, revisiting the treatments, new generation treatments, for the third millennium, personalized medicine, or disease modifying medicine. The effort done worth a title that talks about the challenges, new understanding and how this will change the way to treat this specific group of patients with specific needs.
Response: This was an invited review and the title was provided to the authors. We agree that the title is somewhat generic, but the focus on current approaches and future directions is consistent with the content provided in the review and so hopefully the reader will know what they are getting in the article with the use of this title.
2. The historic perspective of the new understanding PD not only as a motor disorder / neurological disease but potentially affecting cognition must be indicated, so the current perspective makes evident it is an important emerging field (as compared to classical concept based on its motor symptoms) and more endeavours are needed.
Response: We added the following statement to our introduction, “Parkinson’s disease (PD) was historically classified as a movement disorder, but cognitive impairment is prevalent, especially later in the disease process.” (line 49-51)
3. Differential diagnosis should be specified, as well as the state of art of early markers of premorbid and prodromal stages (i.e. sensory, biological, emotional, neuroimage,..) as they play a key role to start interventions. This is mentioned in a small sentence somewhere else.
Response: The focus of this paper is the treatment of cognitive impairment in PD and not the diagnosis of PD-MCI or PDD, as they are covered in another article of the same special edition of the journal. We provided a very brief introduction on the concepts of PD-MCI and PDD so that this treatment article can stand alone, but a more in-depth discussion is reserved for the matching articles.
4. Sleep disorder is not mentioned in the introduction despite the implications in QoL of patients and caregivers and one must wait to line 143 to find it for the first time. Since it is well developed later, at least some early mention in the introduction would be needed.
Response: While we appreciate that sleep can relate to cognition and that many non-motor symptoms overlap, given the amount of information covered in this review, we focus exclusively on treatments for cognitive symptoms and do not address treatments for other (even related) non-motor symptoms. We mention briefly at the beginning of Section 2 that symptom management (e.g. of mood disorders, behavioral disturbance, sleep disorder, etc) is part of the current approach and we re-mention this briefly in section 2.3 (page 12). We also note in section 2.3 that examining treatments for this related symptoms is outside the scope of the review: “A plethora of non-cognitive symptoms can manifest in PD-MCI and PDD, including fatigue, depression, anxiety, apathy, psychosis, REM sleep behavioral disorder, insomnia, impulse control disorder. Treatments for these symptoms are a key part of management of cognitive impairment in PD, but are beyond the scope of this review. Clinicians should screen for non-motor symptoms accompanying cognitive impairment in PD and treat with phar-macologic and non-pharmacologic strategies as needed.” (page 13, line 503-509).
5. 86 male sex. Authors should take into advantage of this sex/gender major issue and introduce and/or discuss its implications in gender medicine. Are the drug pharmacokinetics, pharmacodynamics, side effects, etc the same for males and females with PD disease? To which extent prevention/onset/severity treatments differ?
Response: While the reviewer raises interesting issues, there is not currently good information regarding gender implications for most diseases, including Parkinson disease. While PD is more common in men than women, the biologic basis for this is not well understood. Additionally, there are no differences in treatment approaches for men and women with Parkinson disease. We have added the following sentence to make that point: “While PD is more common in men than women, there currently exists no differences in treatment approaches in men and women with PD.” (page 4, lines 163-165)
6. 94 The underlying mechanism of cognitive impairment in PD is not well 94 understood, and likely includes multiple factors: abnormal protein deposition, loss of dopaminergic neurons, neurotransmitter deficits, synaptic dysfunction, genetics [1], fatty acid oxidation [10], and perhaps inflammation [11,12]. Except for the specific neuronal loss of DA neurons, all the other targets apply for most neurodegenerative disorders, so this sentence should be more specific. Please, keep in mind that the introduction should also provide the state of art of biological targets that will drive the design, repurposing and choice of treatments, which is the final goal of this review. Another example is: 104 There is also significant overlap between PDD and AD pathology, characterized by low CSF levels of amyloid β-42 (Aβ-42) and cortical deposition of Aβ-42 on neuroimaging [11]. Probably, instead of writing such a statement, it would be better to present the same sentence as a challenge or shortcoming for differential diagnosis and its implications with regards to the design or choice of treatment.
Response: As noted above, and as shapes our response to Reviewer #3, the purpose of the review was to focus on treatments for PD-cognitive impairment for the practicing clinicians, also with a brief look to the future, rather than to describe state of the art biological targets to drive design, repurposing, and treatment choice. Thus, we maintain the focus on currently available treatment options while also briefly looking to the future. We have added a sentence that the targets mentioned apply for most neurodegenerative disorders: “Many of these processes are also involved in other neurodegenerative diseases.” (page 3). To the reviewers last comment, we have added the sentence “This overlapping pathology presents as a diagnostic challenge in evaluating individuals with cognitive complaints.” (page 3). We also mention in the new “challenges to study design in PD cognitive impairment” section that deciding whether or not to account for AD pathology is an important consideration in trial design (lines 586-588).
7. 106 PD dementia correlates with Alzheimer’s pathology, and individuals with both pathologies have more rapid disease progression (mean survival of 4.5 years, 108 versus 10 years in those without AD pathology). The sentence is mixing the clinical diagnosis (AD) with the biological conceptual frame referring to BA and tau neuropathologies, which are the hallmarks of AD but not the only ones. Therefore, the statement ‘Alzheimer’s pathology’ includes more neurophysiological derangements than those the authors want to refer.
Response: We have rewritten the sentence to read: “PD dementia correlates with neuropathologies commonly seen in Alzheimer’s disease (amyloid and tau), and individuals with both PD and AD neuropathologies have more rapid disease progression (mean survival of 4.5 years, versus 10 years in those without AD neuropathology)” (page 3).
8. The work would benefit to discuss how ‘Research Domain Criteria (RDoC, NIMH) that is how ‘Dysfunction and dysregulation at the genetic, neural, and behavioral levels’ may fit in this clinical scenario.
Response: Because this review is focused on clinical treatment (see response above, including how we now preface this better in the introduction), we have not reorganized the future research section according to specific domains. Some of Reviewer #4’s suggestions would serve as an excellent article for a journal/review focused more on the neuroscience of cognitive impairment in PD, but are outside the scope of the current review.
9. Table 1. Do the authors consider that psychosis does not imply cognitive changes, so they can go together in a same description as different nosology entities? Please, reconsider the description.
Response: We have deleted “psychosis” from the table title. It now reads “Neurotransmitters Targeted for Pharmacologic Treatments for Parkinson Disease Cognitive Changes.” We agree that the current review focuses on cognition rather than on non-motor symptoms that can accompany cognition changes in PD.
10. Considering that the table is aimed to depict the link between targets and design/use of pharmacological treatments, the descriptive “Medication or medication family targeting” (this implies also posology, etc) should better be referred as “Drugs targeting…”
Response: As suggested, we changed the description to “Drugs targeting cognition in Parkinson disease”.
11. Table 1. Since this review is aimed to provide a historic perspective, past, present and future of treatments, I’d suggest to add this perspective into the tables. For instance, table 1, to add an extra column with the year of implementation of the strategy or drug.
Response: The review is not aiming to provide a historic perspective, which we have now clarified in the introductory paragraphs (lines 55-59). We apologize for the confusion and appreciate the chance to more clearly state the aim of the article.
12. The superfixes a,b,c,d are relevant informations. I’d suggest to highlight them, by including 4 columns, one for each stage, so each drug is placed in one category This will also provide the view of the different stages/purposes of drugs. This would be similar to what is done in Table 2
Response: We added two more columns designated for the drugs’ status for use in PD-MCI and PDD as suggested.
13. Figure 1 on the approach should go before Table 1.
Response: In reframing the flow of the review in response to multiple reviewers’ comments, we have maintained Figure 1 occurring after Table 1. Table 1 accompanies the background concept that cognitive impairment in PD may relate to different mechanisms and neurotransmitters (Section 1). Because Figure 1 outlines treatment approaches and reflects the results in Section 2, we have maintained Figure 1 in Section 2. We felt that this best fit Reviewer 3’s recommendations on flow.
14. Figure 1. ‘medications’ should be better replaced by drugs or treatments (if non-pharmacological ones should also be included)
Response: It now reads “Other drugs tried: other AchEIs, memantine. Other treatments under development”
15. Table 2. Clinically available pharmacologic treatments ‘studied’
Not sure if the verb studied is appropriate here.
Response: We deleted “studied” from the title. It now reads “Table 2. Clinically available pharmacologic treatments for cognition in Parkinson’s disease mild cognitive impairment and Parkinson’s disease dementia”
16. Treating cognitive changes in PD starts with an assessment for reversible 158 contributors to cognitive changes and addressing potential contributors.
Response: We have deleted the second half of the sentence due to redundancy. It now reads “Treating cognitive changes in PD starts with an assessment for potentially reversible contributors to cognitive changes”.
Please, note redundance that can be improved.
17. 162 and 163 … ‘psychosis and cognitive changes’ . Please, be aware that psychosis involves changes in cognition, so the sentence should be amended in the way suits better to the authors.
Response: We deleted “psychosis”. It now reads “Medications used to treat motor symptoms in PD can also contribute to cognitive symptoms”.
18. First generation antihistamines Mind the gap
Response: We changed the alignment to “align left.” Thank you.
19. 176-184 and also 425 - Non-pharmacological treatments are under-represented (only well addressed for PD-MCI) if we consider what the different international PD societies and family associations say to this respect or they do when organizing an important number of their activities to improve emotionality, cognition and QoL of PD patients and caregivers/families independently of the severity of the stage (i.e. dance therapy, etc).
Response: To develop this review, we identified studies (pharmacologic and non-pharmacologic) attempting to treat cognitive outcomes specifically. We agree with the reviewer that we found more studies looking at non-pharmacologic approaches for PD-MCI than for PD dementia. We agree that non-traditional treatments may be underrepresented in research, particularly for PDD, but we included the ones that we found. We have edited the first sentence of this section to read, “Studies on the effect of non-pharmacologic approaches on cognition in individuals with PDD are scarce” to emphasize that the lack of presented data relates to the lack of identified studies, not biases on the part of the review. We also highlight this lack in the new section “Clinical Treatment Approaches for PDD,” where we note, “Research on non-pharmacologic approaches for PDD are lacking…” And we also highlight this in the new conclusion where we say “Current research approaches include both drug repurposing and identifying drugs with novel mechanisms of action. Additional non-pharmacological intervention studies are also needed, especially for PD dementia, where such research is particularly lacking” (page 19). We agree that non-pharmacologic approaches may be important in the overall quality of life in individuals with Parkinson’s with cognitive impairment by treating issues such as mood. However, the focus of this review was on the treatment of cognition, not emotion/mood, other non-motor symptoms, or quality of life.
Please, also revise this statement “Studies on the effect of exercise on cognition in individuals with PDD 426 are scarce. One systemic review examined the effect of exercise in LBD included 5 studies and only 3 included individuals with PDD (total of 10 individuals with PDD) [52].”
Response: We revised to say, “Studies on the effect of exercise on cognition in individuals with PDD 426 are scarce. One systemic review that examined the effect of exercise in LBD included 5 studies and only 3 included individuals with PDD (total of 10 individuals with PDD) [52].”.
20. Drug repurposing is a common practice in pharmaceutical companies but scarcely addressed or visualized in other research studies. Since the authors’ efforts to summarize the developing research (or emerging fields) in Tables 4 and 5 are a major contribution of the current review and may be a differentiation of previous reviews on the approaches and directions, I’d suggest to emphasize it in subheading 3. Developing research, and instead highlight the three novel strategies: 3. Drug repurposing, novel medications and non-pharmacologic interventions. In a certain extent, this 3rd part is talking about the immediate future.
Response: We rewrote the introduction in 3. Developing Research to emphasize the 3 types of treatments in development “studies are investigating treatments of cognitive impairment and dementia related to PD, using various approaches, including (1) drug repurposing, (2) novel medications and (3) non-pharmacologic interventions”. We also included further explanation of 3.1 Drug Repurposing and 3.2 Novel Medications after Tables 4 and 5. Additionally, we made a new subheading 3.3 to refer the readers to sections 2.1.2 and 2.2.2. for non-pharmacological treatments for PD-MCI and PDD, respectively.
21. The last sentence of the 4rt part, is clear in the relevance of ‘drug repurposing’ and ‘identifying drugs with novel mechanisms of action’. These could also be interesting for the title: “Classical treatments, drug repurposing and new disease-modifying treatments to counteract the challenge of Parkinson’s disease with cognitive impairment.”
Response: Thank you for the suggestion. However, the title of the paper was assigned to us for this invited review and the suggested title goes beyond the scope of the review (the emphasis is more on current treatments than future directions).
Round 2
Reviewer 3 Report
The manuscript has been improved as suggested.
Reviewer 4 Report
The authors have provided extensive explanations and arguments for each of the long list of questions. Their resubmitted Ms has incorporated those aspects where we agree they new contents could benefit the final version and other have lest behind due to the intrinsic limitations of an invited report for a Special Issue with a pre-determined number of goals.
It is my pleasure to endorse this work in the present work on the day of today, 11 April 2021.